# Dynamic Multiple Object Segmentation with Spatio-Temporal Filtering

**DOI:** 10.3390/s24072094

**Published:** 2024-03-25

**Authors:** Wenguang Yang, Kan Ren, Minjie Wan, Xiaofang Kong, Weixian Qian

**Affiliations:** 1School of Electronic and Optical Engineering, Nanjing University of Science and Technology, Nanjing 210094, China; 429004@njust.edu.cn (W.Y.); k.ren@njust.edu.cn (K.R.); minjiewan1992@njust.edu.cn (M.W.); 2Jiangsu Key Laboratory of Spectral Imaging & Intelligent Sense, Nanjing University of Science and Technology, Nanjing 210094, China; 3National Key Laboratory of Transient Physics, Nanjing University of Science and Technology, Nanjing 210094, China; xiaofangkong@njust.edu.cn

**Keywords:** feature point tracks, multi-object detection, trajectory distinctiveness

## Abstract

This article primarily focuses on the localization and extraction of multiple moving objects in images taken from a moving camera platform, such as image sequences captured by drones. The positions of moving objects in the images are influenced by both the camera’s motion and the movement of the objects themselves, while the background position in the images is related to the camera’s motion. The main objective of this article was to extract all moving objects from the background in an image. We first constructed a motion feature space containing motion distance and direction, to map the trajectories of feature points. Subsequently, we employed a clustering algorithm based on trajectory distinctiveness to differentiate between moving objects and the background, as well as feature points corresponding to different moving objects. The pixels between the feature points were then designated as source points. Within local regions, complete moving objects were segmented by identifying these pixels. We validated the algorithm on some sequences in the Video Verification of Identity (VIVID) program database and compared it with relevant algorithms. The experimental results demonstrated that, in the test sequences when the feature point trajectories exceed 10 frames, there was a significant difference in the feature space between the feature points on the moving objects and those on the background. Correctly classified frames with feature points accounted for 67% of the total frames.The positions of the moving objects in the images were accurately localized, with an average IOU value of 0.76 and an average contour accuracy of 0.57. This indicated that our algorithm effectively localized and segmented the moving objects in images captured by moving cameras.

## 1. Introduction

Object segmentation plays an important role in the field of computer vision. Our primary focus in image processing is to identify the regions of interest within the image. Here, we refer to these regions as the foreground or objects, while the remaining areas are referred to as the background. For an input image, the purpose of a segmentation algorithm is to accurately distinguish the image into foreground and background regions. In most cases, the foreground in an image is considered one or several objects, such as a vehicle.

The application scenario of this technique is image region segmentation. Grayscale-based object segmentation is a common method in image segmentation and utilizes the grayscale information of pixels in an image to segment the object regions. This method typically includes techniques such as thresholding, edge detection, and region growing, which analyze the grayscale levels of pixels to determine the boundaries between the target and background, thus achieving object segmentation. Thresholding and edge-detection-based image segmentation algorithms divide an image into multiple regions. Subsequently, these regions are further classified as foreground or background using other algorithms. On the other hand, region-growing algorithms first identify a portion of the foreground region and then expand this region to obtain the complete foreground area. These fundamental ideas can also be applied to multi-object segmentation. The main difference is that, in addition to distinguishing the foreground and background in the image, further differentiation needs to be performed among the portions attributed to the foreground.

To achieve more precise object segmentation, some research has focused on studying the objects that need to be segmented. For example, in [1,2], the authors focused on pedestrians on the road, whose segmentation objects often occupy large areas in the image, formed by interconnected but different small regions. In [1], the authors focused on the continuity of trajectories in global associations and used a conditional random field (CRF) model to better match the trajectories of objects. Ref. [2] was more focused on predicting the position of the pedestrians first, to reduce the difficulty of object association, thus achieving continuous detection of targets. Objects can also include athletes on a sports field [3]. Compared to pedestrians on the road, this type of object often involves a mutual influence between objects and fast movement speeds. The study of small animals in groups (birds [4,5], bats [6], ants [7], fish [8], cells [9], etc.) is another focus, where the movement of such targets is more difficult to estimate, and there is a higher degree of occlusion between them. Vehicles are also one of the key objects of segmentation that researchers focus on [10]. First, vehicles in images are a very typical type of rigid object, and compared to other objects such as pedestrians or animals, vehicles typically have distinct shapes and structural characteristics. Second, vehicles come in different types and brands, and there is a practical demand for research on vehicle recognition and classification.

Currently, research on object segmentation in images mainly focuses on single-frame images or moving objects in static cameras. However, with the rise of drone technology, identifying segmented objects from image sequences captured by cameras on moving platforms has become more valuable and meaningful. Compared to previous application scenarios, the changing background adds complexity to the distinction between foreground and background in images. In this paper, we aimed to differentiate between the foreground and background in the current frame by analyzing the distinct motion patterns of the background and objects across multiple frames. This allows us to identify a portion of the foreground, which satisfies the preconditions for region-growing algorithms. Compared to edge-based segmentation, feature point tracking and small-scale region-growing algorithm computations involve significantly lower computational overheads, which is advantageous for processing large image sequences.

Aiming to accomplish this task, we constructed detected point tracks using the optical flow method [11]. Then, through point classification, we could distinguish between the foreground (objects) and background, as well as identify different objects. Finally, by using the adjusted object points as centers, the segmentation of each object contour could be achieved. Our method makes the following contributions:(1)The dual-threshold DBSCAN algorithm is used to temporally classify and differentiate moving objects with inconsistent motion.(2)An iterative algorithm based on the sample mean and variance is used to differentiate spatial objects.(3)An improved two-step contour generation algorithm is used to extract the contours of the objects.

The remainder of this paper is organized as follows: An overview of the related work is presented in Section 2. Section 3 presents the feature point extraction algorithm for spatiotemporal clustering and filters the feature points by the plane constraint of feature points. Section 4 shows the experimental results for different data sets. Then, the results are analyzed. Finally, Section 5 provides the conclusions.

## 2. Related Work

This chapter introduces the latest techniques for object segmentation. Traditional video segmentation techniques can generally be divided into unsupervised, semi-supervised, and supervised methods.

Unsupervised Video Segmentation: Unsupervised video segmentation methods do not rely on any annotated information and achieve segmentation by clustering or matching pixels in a video sequence. Including solutions for hierarchical segmentation, temporal superpixel, and super-voxels [12]. These methods mostly rely on a bottom-up approach, starting from low-level features in images or videos and gradually deriving higher-level semantic information and object segmentation results. In [13], the author utilized a voting mechanism based on the similarity of pixels in the image to perform object segmentation and iteratively refine the results. Algorithms at the pixel level also include combining the first-order directional derivative of the facet model [14]. These pixel-based approaches are often used for small object segmentation. However, it is important to note that the performance of pixel-based methods for small object segmentation can be influenced by various factors, such as the resolution of the image, the quality of object boundaries, and the level of noise present in the image. Therefore, it is necessary to carefully design and optimize the algorithm parameters to achieve accurate and robust segmentation results for small objects. For unsupervised large-scale objects, motion characteristics play an important role in object segmentation. Large-scale objects often exhibit significant motion characteristics, such as movement or complex deformations, which can be utilized to locate and segment the objects in image sequences. This also applies to the requirements for long-term tracking of objects. In [11], an optical-flow-feature fusion-based video object detection method was proposed with the consideration of temporal coherence among video frames to solve the occlusion problem. In terms of multi-object tracking, such as [15], a method for monitoring multiple object cow ruminant behavior based on an optical flow method and an inter-frame difference method was proposed. However, the method was only suitable for short-term and small-scale motion tracking.In [16], the authors detected moving objects and performed blob analysis. Then, a Kalman filter or particle filter was applied to every object to obtain a predicted position, using Munkres’ version of the Hungarian algorithm to distinguish objects. However, the method was greatly influenced by the motion model.

In semi-supervised object segmentation, only a small portion of the image data are manually labeled as an object or the background. In this scenario, the methods utilize the labeled data to guide the entire object segmentation process. The core idea of semi-supervised methods is to learn the features of the object and background based on the known label information. Graph cuts or conditional random fields are used to minimize an energy function or maximize a probability model for segmentation is the common approach. In [12], the authors employed a modified version of the density peak clustering algorithm to generate super-trajectories. Trajectories were generated starting from known objects in the first frame. To enhance the reliability of the trajectories, a reverse tracking strategy was introduced to counteract the impact of slight camera movements. In [17], the authors performed segmentation in “bilateral space”, which is a high-dimensional feature space [18], and this method segments images by using interpolation and locally connected labeling in bilateral space to label different pixels.

Supervised methods require significant human intervention [19,20]. The traditional supervision method needs to correct the image sequence frame by frame, to obtain a high-quality boundary. This huge workload makes the supervised approach more suitable for specific scenarios such as video post-production.

Deep learning algorithms can be considered a special type of fully supervised algorithm, because they typically require a large amount of labeled data for training. In [21], the authors focused on the interactions of objects and built a spatial graph transformer encoder layer, a temporal transformer encoder layer, and a spatial transformer decoder layer to model the spatial-temporal relationships of the object. In [22]. The authors reduced the number of model parameters by introducing convolutional operators. They then addressed overfitting and other training issues by establishing a compact sample distribution model and modifying the update strategy. In [23], the authors used a high-capacity convolutional neural network for object localization and segmentation. To address the issue of scarce training data, they employed auxiliary tasks to pre-train the model and fine-tune the training results based on actual detection. In Ref. [24], the authors proposed a biomimetic retinal neural network based on neurodynamics time filtering and multi-form two-dimensional spatial Gabor filtering. By using two spatial filters that are orthogonal to each other,  movement direction can be accurately estimated. In [25], the authors combined CenterNet with object recognition to enhance the localization and identification of objects, through a series of detailed designs to reduce ambiguity in network training. It is important to note that, while deep learning algorithms excel in fully supervised settings, they also have some limitations. For instance, in the case of object segmentation tasks, acquiring pixel-level labeled data can be challenging, leading to issues such as data imbalance, labeling errors, and boundary ambiguity.

## 3. Algorithm Scheme

In related research on object segmentation, the definition and extraction of the objects was the initial consideration. In this paper, the point tracks in the image are defined and the motion information in the image is determined by the motion information of the multi-frame images.

We first extract the feature points of the image using a curvature model and describe them with a SURF descriptor [26]. In the next frame, the optical flow method [11] is used to associate the same feature point, from which tracks are constructed. The trajectories represent the temporal correlation of the video sequence. Then, using the motion consistency of the feature points on the steel body, the tracks of the feature points belonging to the same object are distinguished from the other tracks. Finally, the points on an object are used as reference points to obtain the full contour of the object. An overall algorithm flow chart is shown in Figure 1.

### 3.1. Image Initialization and Feature Point Detection

The curvature of a curve reflects the degree of curvature at each point. For a point on a continuous curve, the curvature at this point can be calculated according to Equation (Equation 1)
(1)$$\kappa =\frac{\left|y^{\prime \prime }\right|}{{\left(1+y^{\prime 2}\right)}^{\frac{3}{2}}},$$
where $y^{\prime \prime }$ represents the second derivative of the point *M*, $y^{\prime }$ represents the first derivative of the point, and κ represents the curvature of a point on the curve. The image is a discrete signal, which can be described by the Facet model [14]. Based on this model, the first derivative and second derivative of the image can be calculated. In this paper, the feature points are matched between adjacent frames. Therefore, feature points need to be described. The SURF descriptor was chosen because of its repeatability, distinctiveness, and robustness.

### 3.2. Trajectory Correlation

To record the trajectories, we construct the eigenvector $F_{i}^{frame}=\left(v_{i}^{frame},L_{i}^{frame},\theta_{i}^{frame}\right)$. $v_{i}^{frame}$ is the velocity, $F_{i}^{frame}$ stands for the distance in the *x* and *y* directions, and $\theta_{i}^{frame}$ represents the angle of the point’s trajectory. These can be calculated as follows:(2)$$v_{i}^{frame}=p_{i}^{frame}-p_{i}^{frame-1},$$
(3)$$L_{i}^{frame}=\sum_{num=refer+1}^{frame}\left|p_{i}^{num}-p_{i}^{num-1}\right|,$$
(4)$$\theta =\frac{L_{i}^{frame}\left(y\right)}{L_{i}^{frame}\left(x\right)},$$
where *i* is the label of feature points, and frame is the frame number. $p_{i}^{num}$ represents the image coordinate of the $i_{th}$ feature point. refer is the reference frame and the initial value is 1. This value changes based on the trajectory update. $L_{i}^{frmae}$ can be regarded as the integral of the $v_{i}^{frame}$’s absolute value from the reference frame to the current frame.

According to Equations (Equation 2) and (Equation 3), the speed and distance of all points are shown in Figure 2.

In Figure 2, the red lines represent the feature points located within the objects. As seen, the object points are not obvious in the background points in Figure 2. Upon integration, the contrast between the moving objects and the background becomes more pronounced.

Compared to the speed, the distance of the trajectory is more significant. To distinguish the object from the background,  polar coordinates are used to represent the feature points. The parameter $\left(L,\theta \right)$ expression of the trajectory is shown in Figure 3.

The method of polar coordinates includes the moving distance and direction of the points, which is more conducive to the subsequent clustering.

### 3.3. Temporal Clustering and Plane Constraint

Although the disparity in each trajectory’s distance at the background feature points is large, the density is also large. This makes the points in the background different from the points on the objects. For multiple objects, it should be taken into account that different objects are not consistent with each other. Thus, it is not possible to simply set the number of clusters to two. Therefore, density-based spatial clustering of applications with noise (DBSCAN) was chosen as the basis for clustering the distance data [27].

DBSCAN relies on a threshold between data. When the distance between two elements in the data set is less than the threshold ε, the two elements are considered density-reachable. All density-reachable elements are considered as part of the same cluster. When the data are one-dimensional, the algorithmic is as shown in Figure 4.

In this paper, the clustering threshold between moving objects needed to be different from the clustering threshold between objects and the background. So, the double threshold DBSCAN algorithm was used for the temporal clustering of data.

According to the Algorithm 1 and Figure 5, the entire processing procedure is as follows:
**Algorithm 1:** Process of temporal clustering and plane constraint**Input:** all feature points**parameter:**  1: $thr_{1}$ = smaller clustering threshold  2: $thr_{2}$ = larger clustering threshold**Output:** the clusters of feature points representing different motion modes**Do:**  1: distinguish the points with different movement patterns using $thr_{1}$.  2: use the background points to calculate the plane constraints.  3: use $thr_{2}$ to re-cluster the non-background feature points and distinguish the objects of different motion modes.

Step 1: Try to distinguish the objects and the background first. Since the movements of the objects are unpredictable, a smaller threshold is used to distinguish the objects from the background. Then, mark the cluster with the largest amount of data as the background cluster:(5)$${\mathrm{susTarget}}_{index_{i}}^{frame}=\left\{\begin{array}{c} \mathrm{true}\;p_{index_{i}}\;\mathrm{not}\;\mathrm{in}\;\mathrm{largest}\;\mathrm{cluster}\\ \mathrm{false}\;p_{index_{i}}\;\mathrm{in}\;\mathrm{largest}\;\mathrm{cluster} \end{array}\right..$$

Step 2: Use the background points to calculate the plane constraints.

Since the background points cannot be completely segmented by the clustering algorithm, an additional plane constraint is required for the classification of feature points. The projection between planes satisfies the homography change. So, the points’ coordinates on the ground satisfy the homography change under the overhead perspective:(6)$$p^{frame}=Hp^{ref}=\left[\begin{array}{ccc} H_{11} & H_{12} & H_{13}\\ H_{21} & H_{22} & H_{23}\\ H_{31} & H_{32} & H_{33} \end{array}\right]P^{ref},$$
where $p^{ref}$ are the feature points of the reference frame, and $p^{frame}$ are the feature points of the $frame_{th}$. Equation (Equation 6) can be rewritten as
(7)$$s^{frame}=p^{frame}-p^{ref}=\left[H-I\right]p^{ref},$$
where $s^{frame}$ indicates the distance from the reference frame to the $frame_{th}$. *I* is a 3 by 3 identity matrix. This can be obtained from Equation (Equation 7): (8)$$\begin{array}{c} s_{x}^{frame}=a_{1}p_{x}^{ref}+b_{1}p_{y}^{ref}+c_{1}, \end{array}$$(9)$$\begin{array}{c} s_{y}^{frame}=a_{2}p_{x}^{ref}+b_{2}p_{y}^{ref}+c_{2}, \end{array}$$
where $s_{x}^{frame}$ and $s_{y}^{frame}$ respectively represent the component of *x* and *y*. So, $S_{x,\mathrm{isTarget}}^{frame}=\left[\begin{array}{ccc} p_{x}^{ref} & p_{y}^{ref} & s_{x}^{frame} \end{array}\right]$ and $S_{y,\mathrm{isTarget}}^{frame}=\left[\begin{array}{ccc} p_{x}^{ref} & p_{y}^{ref} & s_{y}^{frame} \end{array}\right]$ form two plane equations $P_{x}^{frame}$ and $P_{y}^{frame}$. For a spatial point set, only three points that are not co-linear are needed to determine the plane. Then, the points that are not judged as background are judged again. The points that satisfy the plane equation in both directions are also considered to be in the background.
(10)$${\mathrm{isTarget}}_{\mathrm{inde}x_{i}}^{frame}=\left\{\begin{array}{c} {\mathrm{true}\;\mathrm{if}\;S}_{x,\mathrm{inde}x_{i}}^{frame}\in P_{x}^{frame}{\;\mathrm{and}\;S}_{y,\mathrm{inde}x_{i}}^{frame}\in P_{y}^{frame}\\ \mathrm{false}\;\mathrm{else} \end{array}\right..$$

Step 3: A large threshold $thr_{2}$ is used to distinguish different objects. For multiple objects, different motion modes produce different motion eigenvalues.

### 3.4. Spatial Point Clustering

When the motion modes of the objects are consistent, which happens in many scenarios, spatial clustering is necessary, as the number of objects cannot be determined and the position relationship of different objects is also uncertain. To better distinguish different objects, we use an iterative algorithm based on the sample mean and variance.

The main idea of clustering is to solve the clustering center and regroup all the data according to the Euclidean distance between the data and the clustering center. When the variance $\left(std_{x},std_{y}\right)$ of a class with a center point (x,y) is greater than the threshold $\theta_{s}$, the cluster is divided. If $std_{x}$ is greater than $std_{y}$, the new center points $(x^{+},y)$ and $\left(x^{-},y\right)$ after splitting are
(11)$$\begin{array}{c} x^{+}=x+s\cdot std_{x}, \end{array}$$
(12)$$\begin{array}{c} x^{-}=x-s\cdot std_{x}, \end{array}$$
where *s* is the scaling factor, which is generally 0.5. If  $std_{y}$ is greater than $std_{x}$, the value *y* changes in the same way. After splitting, the cluster is re-clustered and the new cluster center is calculated.

When there are two clusters whose center distance is less than the threshold, the two clusters are merged. In the iteration process, if the cluster center changes and the number of iterations is less than the maximum iterations, the iteration continues. The algorithm flow is shown in the Algorithm 2.
**Algorithm 2:** Process of spatial clustering**Input:** object points sets**parameter:**  1: κ = the expected number of cluster centers  2: $\theta_{s}$ = the maximum standard deviation of a cluster  3: $\theta_{c}$ = the minimum distance between two cluster centers  4: *I* = maximum iteration times**Output:** space clustering results in the current frame**While:** the clustering results change or the iteration times are less than *I*:  1: Classify according to the cluster center  2: Re-determine the cluster center  3: Determine whether to split a class based on the threshold  4: Re-determine the cluster center  5: Merge the clusters that meet the criteria  6: Re-determine the cluster center

### 3.5. Object Contour Generation

To segment the objects’ contour, this paper segments the object shape based on the detected feature points. We use the combination of point clustering and image matching to judge the label of each pixel around the selected feature points. For points on the plane, the corresponding points between two frames should satisfy the corresponding homography relationship. This can be expressed as
(13)$$p_{frame}=H_{refer}^{frame}p_{refer}.$$

In Equation (Equation 13), $H_{refer}^{frame}$ represents the homography matrix from refer to frame. It is noted that the homography matrix can only constrain points in the background. These points were distinguished in Section 3.3. After obtaining $H_{refer}^{frame}$, the image can be changed to eliminate the influence of the moving platform and obtain the region of interest. This process is shown in Figure 6.

In Figure 6, the generation of the background image adopted the Gauss mixed model. As the number of frames increases, the gray value of each pixel is jointly determined by the gray value of the same position from the reference frame to the current frame. Since the objects in the background are difficult to distinguish, the objects on the reference frame are also projected in Figure 6c. The object area is determined by the extracted object location in Section 3.3. In this case, the selected area will be affected by the reference frame. So, it is necessary to make a further judgment on the pixel points according to the moving points.

In Figure 7, we begin by fusing the points with the region of interest, which is referred to as the first extraction. Then, the marked points on each object are taken as seed points to generate the contour within the region of interest, which is referred to as the second extraction. Finally, the contours of the different objects are generated.

### 3.6. Trajectories Update

With time, the number of trajectory points will inevitably decrease due to the movement of the platform. It is necessary to update the feature points and background images based on preserving trajectory information. Therefore, set the threshold $thr_{num}$ as follows:(14)$$update_{sign}=\left\{\begin{array}{c} true\;\;if\;num_{frame}<thr_{num}\cdot num_{refer}\\ false\;\;else \end{array}\right.,$$
where $num_{frame}$ represents the number of feature point trajectories retained in the current frame. $num_{refer}$ represents the number of feature points detected in the reference frame. The empirical value $thr_{num}$ is 0.5. In addition, feature points are not updated when the trajectory duration is less than 15 frames. When the trajectory information is updated, the accumulated trajectory distance *L* is reset as the reference frame changes. To reduce the influence of the update, a certain initial value is given to the feature points existing in the object area. Then, in the subsequent frames, the feature points on the object can still be distinguished from the feature points on the background through time domain filtering.

## 4. Experiment and Result

This study intended to find multiple moving objects (mainly rigid bodies such as vehicles) under a moving platform. So, we used some sequences in the Video Verification of Identity (VIVID) program database [28] to verify the algorithm. The selected image sequences had interference such as object intersection and object occlusion. Part of the experimental image sequences is shown in Figure 8. Table 1 lists the major challenges of the relevant sequences.

For the algorithm designed in this paper, no preconditions were used except for image sequences. The values of constant parameters are described in Section 3. We used Matlab2018 to implement our method on a computer platform configured with an Intel processor of 3 GHz and 16 gigabytes of RAM.

### 4.1. Moving Point Segmentation

The key point of the algorithm is to determine the number and location of objects according to different movement modes. Therefore, after the tracks have been established, the primary task is to filter the moving points.

Part of the experimental results are shown in Figure 9 and Figure 10.

In the data processing, all images were transformed into gray images. As seen in Figure 9 and Figure 10, the algorithm could identify the feature points belonging to the moving objects from a large number of feature points, and it classified the moving feature points correctly.

We evaluated the model performance using the F1-score: (15)$$\begin{array}{c} P_{i}=\frac{TP}{TP+FP}, \end{array}$$(16)$$\begin{array}{c} R_{i}=\frac{TP}{TP+FN}, \end{array}$$(17)$$\begin{array}{c} F=\frac{2P_{i}R_{i}}{P_{i}+R_{i}}, \end{array}$$
where TP represents the number of points that were correctly classified. FP represents the incorrectly classified points. FN represents the points on the objects that were not detected.

The F-values calculated for the first 300 frames of each sequence are shown in Figure 11. Table 2 shows the proportion of frames that accurately differentiated between moving target points and background points. In Seq. (a), there were a total of 5 trigger detections, and due to its relatively simple background, the F-values for the entire sequence were consistently high. The points on objects and background were well distinguished. The decrease in F-value was mainly caused by the updates of feature points. The overall accuracy of the sequence reached 84%. After the feature points had been updated, the algorithm was able to correctly differentiate between moving feature points and background points within 5 frames. Seq. (b) was similar to Seq. (a), with an overall accuracy of 85%. Multiple objects in Seq. (a) and Seq. (b) were also correctly distinguished from each other. This demonstrated the algorithm’s ability for multi-target segmentation as presented in this study. In Seq. (c), due to the limited number of stable feature points that could be tracked on the background, it was challenging to establish long-term trajectories, resulting in a higher frequency of updates. In the 300 frames of images tested, the feature points were updated 15 times. Additionally, significant camera motion, while the objects themselves exhibited minimal movement after trajectory updates, could have led to a situation where some feature points on the objects were misclassified as background points. In Seq. (d), the overall detection success rate was also 85%. According to Figure 11, there were some frames in the sequence where F-values were missing. Comparing the data, during these frames, the camera focal length was changing, causing an overall blurring of the images and significantly impacting the detection of feature points, failing to detect feature points on the objects. After restoring the image quality, the detection performance returned to normal, demonstrating the robustness of the algorithm. In Seq. (e), which contained large-scale vehicles as objects, the overall accuracy was only 52%. Upon analyzing the classification of feature points, we found that this was due to a significant overlap between the objects and the background. In these overlapping regions, it was easier for feature points to be generated. However, when counting such feature points, we did not consider them as correct feature points on the objects, since their trajectories were unstable. This was the main reason for the significant curvature in the F-value. Due to the complexity of the background, the number of feature points on the background remained stable, leading to only four updates of the feature points. Seq. (f) had the main challenges of intense camera movement and rapid changes in target scale, affecting the stability of the trajectories of both object and background feature points. This also resulted in a faster update of feature points. In the 300 frames of images, the feature points were updated 16 times, which was similar to Seq. (c). This was the reason for the overall accuracy of 75%.

In general, after the trajectory disappears, the updating strategy may cause fluctuations in point classification in a short period. Then, it quickly adjusts and achieves good results. The experimental results indicated that the algorithm proposed in this study performed well in handling occlusion in image sequences. In the case of multiple target vehicles in an image, the algorithm could effectively differentiate feature points belonging to different targets based on their trajectories.

### 4.2. Split Results of the Test Dataset

Finding points on the objects showed the positioning function. Building upon this, we further addressed the target segmentation issue through background differencing and seed generation algorithms. In object segmentation algorithms, there are two commonly used evaluation metrics: the intersection-over-union metric (Y), and the contour accuracy (*F*) [12].

The intersection-over-union score is defined as
(18)$$Y=\frac{1}{n}\left(\sum_{i=1}^{n}\frac{M_{i}\cap G_{i}}{M_{i}\cup G_{i}}\right),$$
where $M_{i}$ is the segmentation mask of the $i_{th}$ object, and $G_{i}$ is the corresponding ground-true mask. *n* is the number of objects.

The contour accuracy *F* is for measuring how well the segment contours match the ground-truth contour. Contour-based precision $P_{i}$ and recall $R_{i}$ between $c\left(M\right)$ and $c\left(G\right)$ can be calculated as follows: (19)$$\begin{array}{c} P_{i}=\frac{TP}{TP+FP}, \end{array}$$(20)$$\begin{array}{c} R_{i}=\frac{TP}{TP+FN}, \end{array}$$
As in Equations (Equation 15) and (Equation 17), TP indicates that both the measured value and the true value are object boundaries. FP represents the part that was incorrectly detected as the boundary. FN represents the undetected boundary.

Then contour accuracy *F* can be defined as
(21)$$F=\frac{1}{n}\sum_{i=1}^{n}\frac{2P_{i}R_{i}}{P_{i}+R_{i}}.$$

Qualitative segmentation results for the sequences are presented in Figure 12 and Figure 13.

As shown in Table 3, Figure 12 and Figure 13, in the case where the overall size of the objects was small, the algorithm in this paper could effectively recognize multiple objects and extract their contours. In some sequences, the contours of the objects were not well segmented, due to the objects being similar to the background, and with the parameters being set consistently.

In Seq. (a), the results demonstrated that the algorithm could effectively segment the objects. The errors mainly stemmed from the shadows formed by the vehicles on the road. Since the shadows moved along with the vehicles, they also complied with the definition of moving targets in the images. For Seq. (b), when there was an intersection between objects, the algorithm could effectively differentiate between different vehicle objects due to their distinct trajectories. However, during the process of intersection, it may not be possible to accurately separate the portions occupied by each object. The segmentation results for Seq. (c) were lower than those of Seq. (a) and Seq. (b). This is because both background differencing and seed generation rely on the differences between the objects and the background. In Seq. (c), the difference between the objects and the background was minimal. Even if feature points on the object can be located, this may not effectively segment a distinct region of the object. For Seq. (d), the segmentation of the target was quite good, with an IoU value of 0.83. When the target is heavily occluded, the segmentation of the target may be greatly disturbed. Then, when the object reappears, it can still be completely segmented. In Seq. (e), the segmentation of the target also depended primarily on the contrast between the target and the background. Additionally, due to the larger size of the target, it became more challenging to achieve complete segmentation. This was the main reason for the lower IoU value and accuracy compared to the other test sequences, which were only 0.62 and 0.49. In Seq. (f), the significant scale change of the object was the main reason affecting the segmentation results. When the scale was small, the IoU value could reach 0.83. However, when the target scale rapidly increased, this often led to the rapid disappearance of feature points as well. The trajectories of feature points on the target were also affected, resulting in unstable segmentation results.

### 4.3. Evaluation on a Youtube-Object

To facilitate comparisons with other algorithms, we ran our algorithm on Youtube-Object sets [29] and compared it with other algorithms: BVS [17], VSF [30], OFL [31], and STV [12]. Since some code was not available, we could compare the sequences reported in papers. The IoU scores of our and the compared methods are presented in Table 4.

In Table 4, the characteristics of the different image sequences are shown. In the aeroplane sequence, the background remained relatively static, which means that the trajectories of background feature points had a higher level of similarity. On the other hand, there was minimal interference from the sky background, which facilitated object segmentation in this case. For the bird sequence, due to the movement of the bird being concentrated on its head, the feature points on its body were classified as background points. This was the reason for the lower IoU value achieved by the algorithm in this study. The algorithm in this study achieved good results when detecting targets that moved as a whole, such as cars and horses. For the train sequence, as the train gradually approached the camera, the algorithm in this study could effectively segment the object. However, when the train got too close to the camera and occupied a large portion of the image, it no longer adhered to the assumption that there were more background feature points than object feature points. As a result, the algorithm failed to segment the target effectively.

## 5. Summary and Future Work

This paper introduced an algorithm that focused on detecting all moving objects in a dynamic background. The algorithm in this paper is based on feature points, establishing trajectories for detected feature points in the image. By utilizing the density and consistency of background points, the trajectories of feature points on the objects are filtered. The algorithm leverages the differences in background point matching to identify foreground objects and uses motion consistency to differentiate between different objects, enabling the detection of multiple objects within the same frame. Through experiments and evaluations on various image sequences, we validated the feasibility of the proposed method.

For the algorithm designed in this study, there were three main factors that influenced the final results: the formation of feature points and trajectory generation, extracting information from trajectories to distinguish between motion feature points and background feature points, and complete classification and segmentation of objects based on the detected object feature points.

The results of this study demonstrate that trajectory classification can effectively differentiate moving objects in a dynamic background. The algorithm performed well in handling issues such as occlusion and sudden changes in image sequences. When the motion of the background in the images became smoother, the algorithm produced good results. Through comparing the results of various experiments, it was found that image feature point detection exhibited more randomness, and the stability of trajectories needs to be improved. On the other hand, segmentation algorithms based on feature points are limited by the positions of the feature points on the objects. The way feature points are generated determines that they are more likely to appear at the edges of objects. Although this study selected points by considering the positions of different points on the same object comprehensively and prioritized selecting intermediate points to increase the probability of seed points being inside the object, there was still room for improvement in the overall segmentation performance.

Therefore, in future research, we aim to focus on improving trajectory stability and object segmentation by using feature line trajectories in images, instead of the current feature points. Feature lines can provide more stable feature descriptions, helping to form more stable trajectories. When an object is surrounded by multiple sets of feature lines, the likelihood of obtaining its complete outline increases. Additionally, dealing with the trajectory classification issue brought about by sparse feature lines on the background compared to feature points will also be addressed.

## Figures and Tables

**Figure 1 sensors-24-02094-f001:**
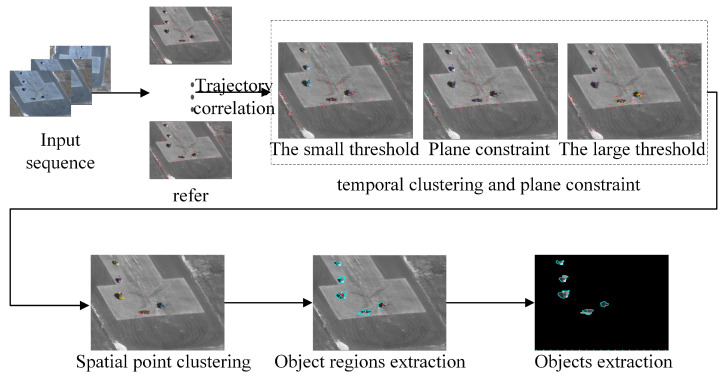
Overall algorithm flow chart.

**Figure 2 sensors-24-02094-f002:**
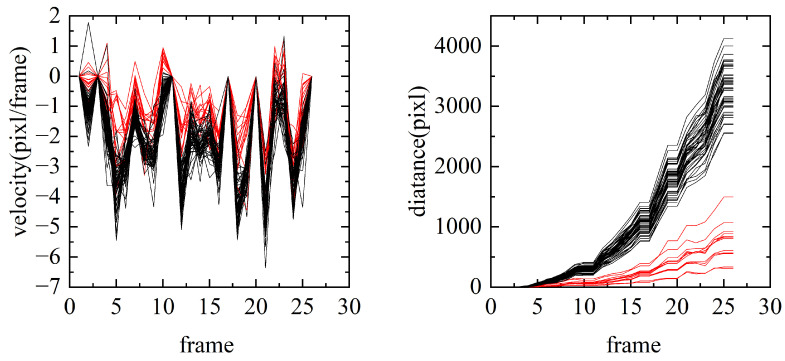
The velocity and distance representation of the trajectories in one direction. The X-axis represents the frame number.

**Figure 3 sensors-24-02094-f003:**
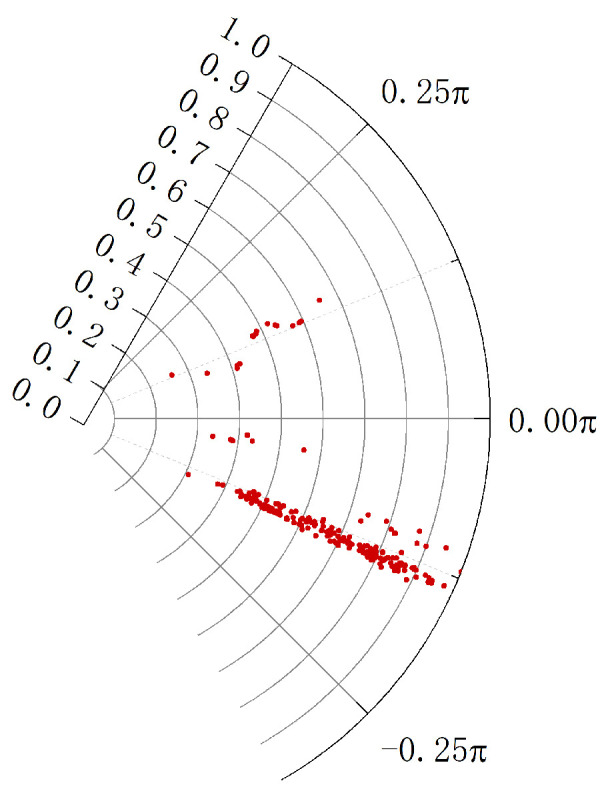
Polar representation of the trajectory of feature points in a frame.

**Figure 4 sensors-24-02094-f004:**
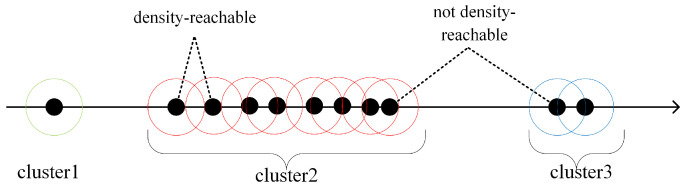
Density clustering. All data are divided into three clusters, and elements in each cluster are less than a threshold (radius of the circle) from at least one element in the same cluster.

**Figure 5 sensors-24-02094-f005:**
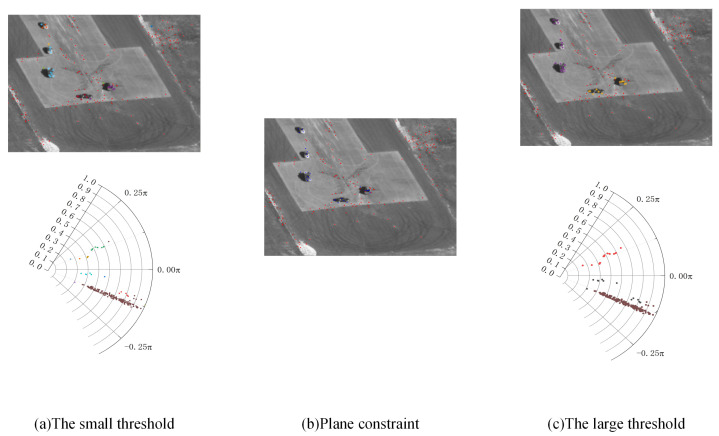
(**a**) Using small thresholds to distinguish backgrounds. (**b**) Screening the feature points by plane constraints. (**c**) Using large thresholds to distinguish backgrounds.

**Figure 6 sensors-24-02094-f006:**
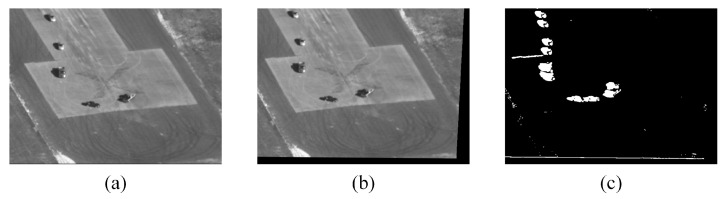
The image is aligned by a homography matrix. (**a**) represents the generated background image $I_{1}$, (**b**) represents the image $I_{reback}$ after changes through the homography matrix, and (**c**) represents the gray difference between the current frame and the background frame after binarization.

**Figure 7 sensors-24-02094-f007:**
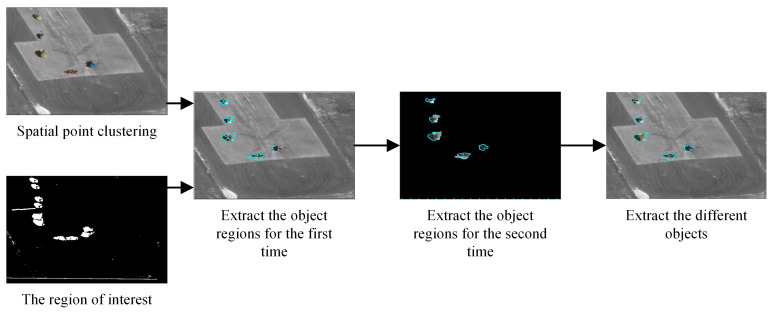
Extraction of objects based on background points and background regions.

**Figure 8 sensors-24-02094-f008:**
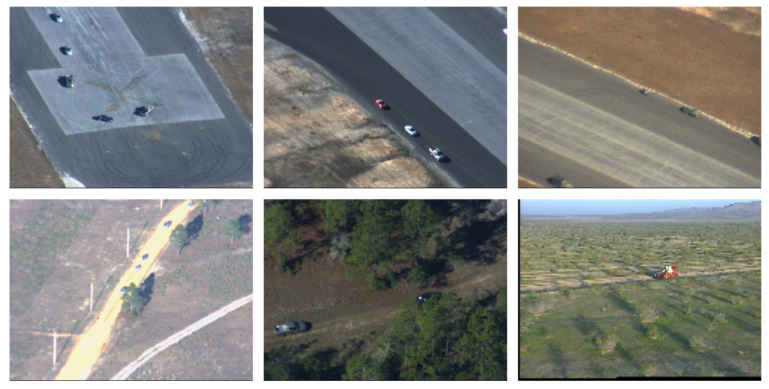
Part of the experimental image sequences.

**Figure 9 sensors-24-02094-f009:**
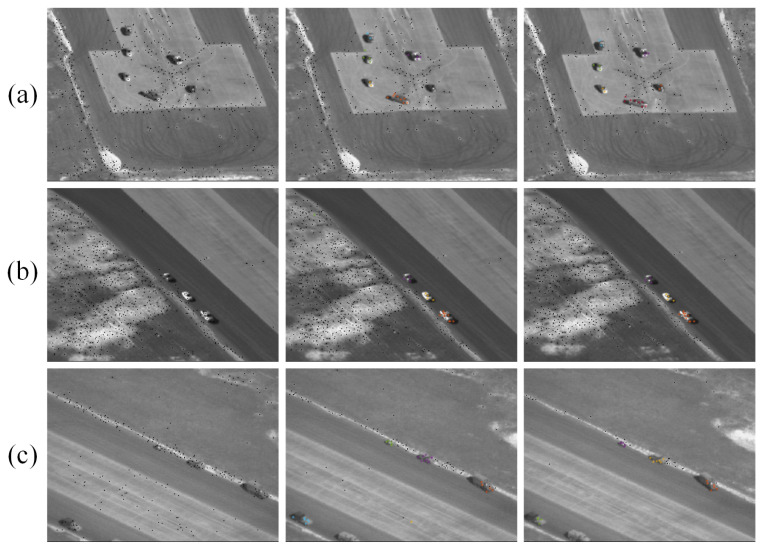
Motion feature points screening (Seq. (**a**)~Seq. (**c**)).

**Figure 10 sensors-24-02094-f010:**
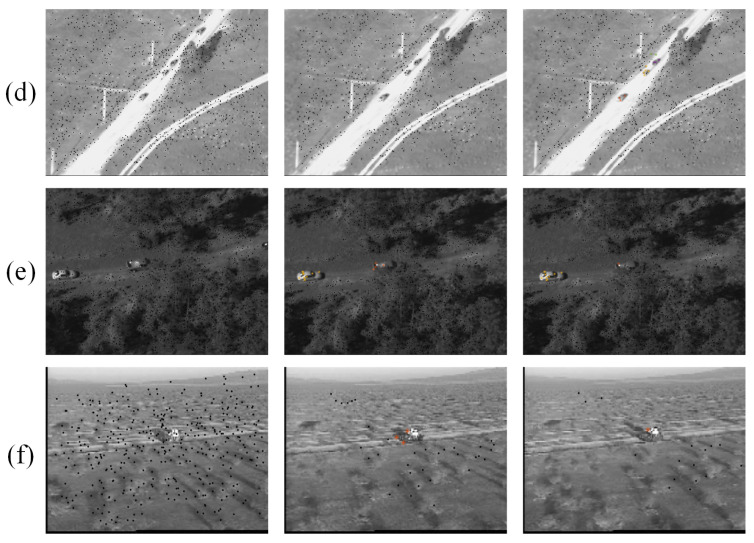
Motion feature points screening (Seq. (**d**) ~Seq. (**f**)).

**Figure 11 sensors-24-02094-f011:**
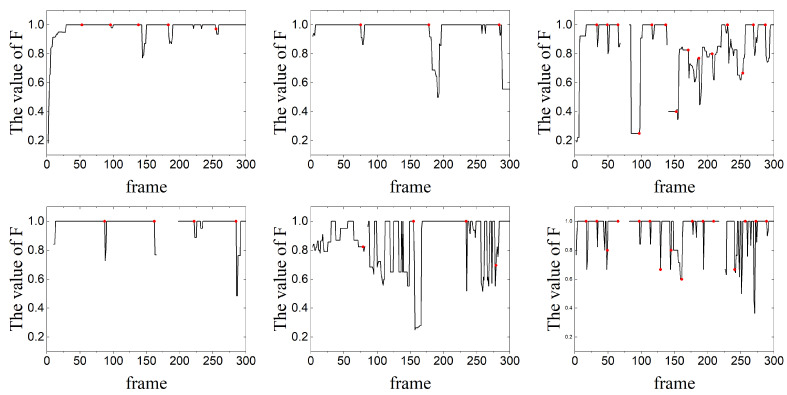
F values obtained from different sequences. The red dots represent the updated frames.

**Figure 12 sensors-24-02094-f012:**
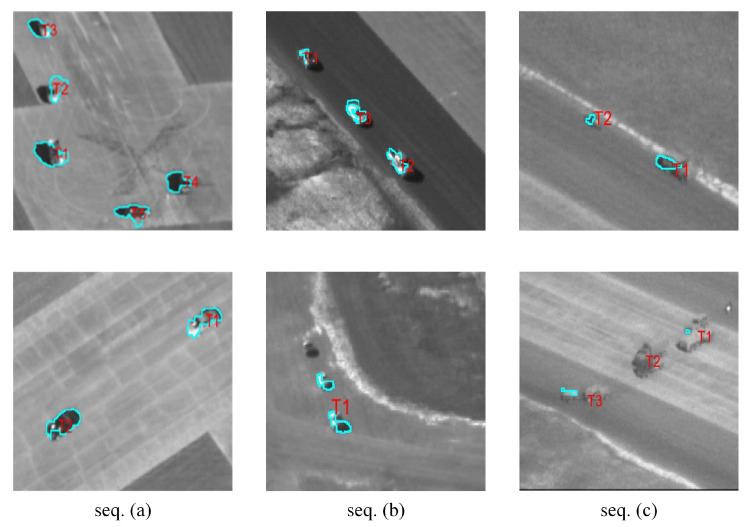
Qualitative segmentation results on Seq. (**a**)~(**c**). The results show that the proposed algorithm could cut out multiple objects well and distinguish different objects.

**Figure 13 sensors-24-02094-f013:**
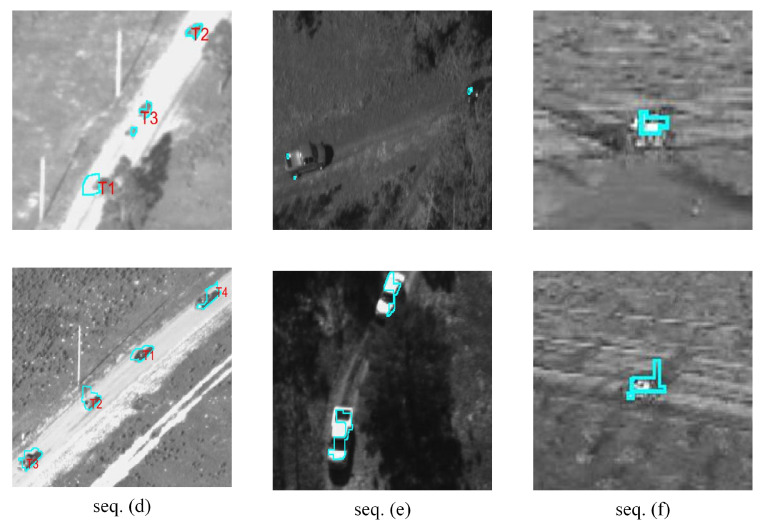
Qualitative segmentation results on sequences. (**d**)~(**f**). The results show that the proposed algorithm could cut out multiple objects well and distinguish different objects.

**Table 1 sensors-24-02094-t001:** Introducing the existing problems of each image sequence.

Seq.	Challenge
(a)	There are more moving objects.
(b)	The objects are close together.
(c)	The objects’ contrasts are not high.
(d)	The image is blurred and there are occlusion.
(e)	Interference is bigger and a large number of background points are not on the ground.
(f)	There are scale variations.

**Table 2 sensors-24-02094-t002:** The accuracy of detection points for each sequence.

Evaluation Tools	Seq. (a)	Seq. (b)	Seq. (c)	Seq. (d)	Seq. (e)	Seq. (f)
Accuracy	84%	85%	40%	85%	52%	75%

**Table 3 sensors-24-02094-t003:** IoU score (Y) and contour accuracy (*F*) averaged value on the tested sequences. Higher values of IoU score and contour accuracy are better.

Evaluation Tools	Seq. (a)	Seq. (b)	Seq. (c)	Seq. (d)	Seq. (e)	Seq. (f)
IoU	0.86	0.83	0.75	0.83	0.62	0.65
contour accuracy	0.64	0.63	0.58	0.66	0.49	0.42

**Table 4 sensors-24-02094-t004:** IoU score on the some Youtube-Objects dataset. Higher values mean better results.

Dataset	Category	BVS	VSF	OFL	STV	OURS
Youtube-Object	Aeroplane	0.808	**0.89**	0.853	0.811	0.83
	bird	0.764	0.816	**0.831**	0.813	0.62
	car	0.567	0.709	0.606	**0.754**	0.72
	horse	0.531	0.678	0.623	0.716	**0.73**
	train	0.621	**0.782**	0.747	0.761	0.621

Bold numbers indicate the best-performing values across all algorithms.

## Data Availability

Dataset available on request from the authors.

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
