# Peer review of "Dynamic Multiple Object Segmentation with Spatio-Temporal Filtering"

_sensors, 2024, doi:10.3390/s24072094_

Round 1

Reviewer 1 Report

Comments and Suggestions for Authors

The article addresses a very interesting topic.

You could try to improve the exposure. Sometimes it is hard to follow. In any case, even as it is now it is easy to understand. There are only a few things that I think should be fixed. First of all, the figures.

Figure1, Figure2, Figure5, Figure7 and Figure11 have the labels too small.

Especially the graphs in Figure5 and Figure11 are not readable.

Then in Tbbella 3 in my opinion the best result for each dataset should be highlighted in bold.

I also recommend adding to the references on line 24 the work https://ieeexplore.ieee.org/abstract/document/7062911#full-text-header that led to the publication of some work in the journal nature.

Reviewer 2 Report

Comments and Suggestions for Authors

1. The correspondence email should be an official email address belonging to the affiliated institution.

2. Acknowledgement should be a separate section.

3. The abstract does not provide information about the algorithm's background, the dataset and, most importantly, does not introduce quantitative findings and the significance of these findings.

4. The first keyword is not appropriate.

5. line 23 thre is a typo on wehicles (Last word).

6. The introduction is very superficial. Even with the related works section (which is also short and superficial), the paper does not introduce a good definition of the problem, the history and previous work, the evolution of the methodologies applied for solving the problem, and how this work significantly improves the current methods.

7. Paper detects the movement pattterns with respect to changes in points in the temporal plane, and did not discussed the changing viewing geometry of the acquisition across multitemporal scenes.

8. I could not find any novel part in point clustering and object contour generation.  Especially the contour generation, which is mentioned to be an improved version is not explained well, and also it is not supported by the visuals where the improvement is.

9. The caption of Figure 8 is not appropriate. 

10. Figures are not in an appropriate zoom scale thus not enable understanding the the performance of segmentation.

11. There is a very limited discussion of the findings, limitations and future work. I recommend having a separate discussion section.

12. Similar to the abstract, I recommend putting quantitative findings in context.

Reviewer 3 Report

Comments and Suggestions for Authors

Please see attached PDF.

Comments on the Quality of English Language

Needs major English editing.

Round 2

Reviewer 2 Report

Comments and Suggestions for Authors

Authors performed a good revision that provided answers to all my concerns. Especially the improvements in introduction and results are now at an appropriate level. 

Author Response

Repay: Thank you very much for your recognition of this paper.

We have made a correction to the paper and the modifications in this round are marked in red.

Reviewer 3 Report

Comments and Suggestions for Authors

Revised manuscript looks OK

Comments on the Quality of English Language

English is OK

Author Response

(The authors gave the same response as above.)
